# Parasitic Granulomatous Dermatitis Caused by *Pelodera* spp. in Buffalo on Marajó Island, Pará

**DOI:** 10.3390/ani14091328

**Published:** 2024-04-29

**Authors:** Camila Cordeiro Barbosa, Carlos Eduardo da Silva Ferreira Filho, Carlos Magno Chaves Oliveira, Tatiane Teles Albernaz Ferreira, Marilene de Farias Brito, Stella Maris Pereira de Melo, Milena Carolina Paz, Saulo Petinatti Pavarini, David Driemeier, José Diomedes Barbosa

**Affiliations:** 1Instituto de Medicina Veterinária, Universidade Federal do Pará, Castanhal 68740-970, Brazil; carloseduardofilho.mv@gmail.com (C.E.d.S.F.F.); cmagno@ufpa.br (C.M.C.O.); tatyalbernaz@ufpa.br (T.T.A.F.); diomedes@ufpa.br (J.D.B.); 2Departamento de Epidemiologia e Saúde Pública (DESP), Instituto de Veterinária (IV), Universidade Federal Rural do Rio de Janeiro (UFRRJ), Seropédica 23890-000, Brazil; mfariasbrito@uol.com.br; 3Setor de Patologia Veterinária, Faculdade de Veterinária, Universidade Federal do Rio Grande do Sul (UFRGS), Porto Alegre 91540-000, Brazil; stellamaris1910@hotmail.com (S.M.P.d.M.); mileenapaz@hotmail.com (M.C.P.); sauloppvet@yahoo.com.br (S.P.P.); davetpat@ufrgs.br (D.D.)

**Keywords:** parasitic disease, rhabditoid nematodes, *Bubalus bubalis*, Amazon biome

## Abstract

**Simple Summary:**

Buffalo are country animals, with great adaptability, and thrive in diverse environments. The State of Pará has the largest herd of buffalo in Brazil, which can be found mainly on Marajó Island in the northern region of the state. As a result, buffalo farming has become widespread in Brazil, and it offers several products to the consumer market. However, in the State of Pará, skin diseases stand out among the diseases that affect this species. This study reports the first case of dermatitis caused by *Pelodera* spp. in buffalo (*Bubalus bubalis*) on Marajó Island, Pará.

**Abstract:**

This is the first report of parasitic granulomatous dermatitis caused by *Pelodera* spp. in a buffalo. The affected buffalo was about seven years old, was a female of the Murrah breed and belonged to a property located on Marajó Island in the State of Pará. During the clinical examination, the animal was in a standing position and presented several multifocal nodular and placoid masses throughout the body, mostly on the forelimbs, hindlimbs, abdomen, mammary glands, perineum, vulva and tail. These masses were also observed on the nasal mucosa, head, neck, back and chest. On macroscopic examination, the skin had several multifocal-to-coalescent sessile nodular and placoid lesions. Histopathology of the skin showed a marked reduction in the number of hair follicles. In the superficial dermis, there was significant multifocal-to-coalescent inflammatory infiltration, consisting of macrophages, epithelioid macrophages, lymphocytes, plasma cells and multinucleated giant cells. In the remaining hair follicles, there were numerous cross and longitudinal sections of small rhabditoid nematodes characterized by a thin cuticle, platymyarian musculature, an intestinal tract, a rhabditiform esophagus and lateral alae (morphologically compatible with *Pelodera* spp.). The diagnosis of parasitic dermatitis was confirmed by histopathological skin lesions associated with the presence of intralesional rhabditiform larvae morphologically compatible with *Pelodera* spp.

## 1. Introduction

The State of Pará has the largest herd of buffalo in Brazil, which can be found mainly on Marajó Island in the northern region of the state [1]. These animals have a thick epidermis, a high concentration of melanin in the skin, a small number of sweat glands and few hair follicles [2]. Therefore, there is a theory that these animals are resistant to skin diseases. In the State of Pará, skin diseases such as those caused by parasites, tumors, lesions caused by incorrect handling, hot-iron marking, poxviruses and leucodermia stand out among the diseases that affect this species [3,4].

Among the dermatitis variants of parasitic origin, the most common in buffalo are those caused by lice and scabies [4]. *Pelodera* spp. is a saprophytic parasite with a life cycle in decomposing wet organic matter [5]. Larvae in the L3 stage occasionally invade the skin, penetrating the follicular infundibulum and causing pruriginous dermatitis [6]. Dermatitis caused by *Pelodera* spp. has been reported in dogs [7,8,9], cattle [10,11], horses [12], sheep [13], guinea pigs [14], seals [15], bears [16,17] and humans [18,19]. The affected animals exhibit intense pruritus, and lesions develop in areas that were in contact with surfaces contaminated with the parasite [13,14]

A literature search did not reveal any studies conducted in Brazil reporting the occurrence of *Pelodera* spp. in animals. Moreover, there are no reports of this skin disease in the buffalo species. Thus, the objective of the present report is to describe the first case of dermatitis caused by *Pelodera* spp. in buffalo (*Bubalus bubalis*) on Marajó Island in Pará.

## 2. Case Report

A seven-year-old female buffalo of the Murrah breed, weighing approximately 400 kg, was under observation in a property located in the municipality of Santa Cruz, Marajó Island, Pará (−0.894922936378, −49.0738353788). According to the caretakers, over the previous two years, the animal had gradually lost weight and multiple nodules of varying sizes had appeared over its body. In view of the animal’s clinical presentation, the owner requested a technical visit to the property.

On clinical examination, the animal was standing, in poor nutritional status and had respiratory distress, serosanguineous nose discharge, purulent vaginal discharge, and several multifocal-to-coalescent, sessile nodular and placoid nodular masses throughout its body, predominantly on the forelimbs and hindlimbs, abdomen, mammary gland, perineum, vulva and tail, but also on the nasal mucosa, head, neck, back and chest (Figure 1A,B).

Three young buffaloes and one adult with similar clinical signs were observed on Marajó Island by veterinary colleagues on properties in the same municipality as the animal in the aforementioned study.

In the ultrasound examination of the nodules, ovoid structures were observed in the subcutaneous tissue with a multifocal distribution. They were sometimes coalescent, well defined, sometimes they had a thicker capsule and other times they had a thin hyperechoic line around their perimeter. These structures were composed of heterogeneous material with echogenic areas and hyperechogenic misshapen areas, which sometimes presented hypoechoic fluid tending to anechoic inside (Figure 2A), suggesting that they were granulomas and pyogranulomas, but no signs of mineral deposits were seen in the structures, such as hyperechoic areas followed by an acoustic shadow.

The animal died and necropsy was performed. On macroscopic examination, the skin had several multifocal to coalescent sessile nodular and placoid lesions of varying sizes (10 cm to 15 cm in diameter). On the ventral side of the tail, the lesions were predominantly ulcerated and reddish (Figure 2B).

On the nasal mucosa the lesions were projected into the lumen and partially obstructed the nasal cavity. When cut, the masses were well delimited, soft and yellowish; had a necrotic and caseous center; and were surrounded by a whitish and firm capsule (Figure 3A). After the necropsy, the metacarpals, carpals, metatarsals, tarsals, radius and ulna of the right hindlimb were macerated, and rough-to-irregular osteophytes and concavities/depressions on the diaphysis of the metacarpals were observed on the surface of these bones. X-rays showed periosteal reactions, joint space narrowing and osteolysis, with osteophytes and enthesophytes (Figure 3B,C). Samples were collected from several organs and fixed in 10% formalin for subsequent histological analysis and stained with hematoxylin and eosin. In addition, samples of the skin and lymph nodes were stained with Grocott methenamine silver and Ziehl–Neelsen stain.

Histopathology of the skin showed a marked reduction in the number of hair follicles. In the superficial dermis, there was significant inflammatory infiltration of macrophages, epithelioid macrophages, lymphocytes, plasma cells and multinucleated giant cells. Amidst this infiltration, there were areas of multifocal necrosis with significant deposition of cell debris (Figure 4A,B). In the remaining hair follicles, there were numerous cross and longitudinal sections of small rhabditid nematodes characterized by a thin cuticle, platymyarian musculature, an intestinal tract, a rhabditiform esophagus, and lateral alae (morphologically consistent with *Pelodera* spp.) (Figure 4C,D). In another sample of skin, the dermis also presented nodular areas with significant caseous necrosis, deposition of strongly basophilic material (mineralization) and significant inflammatory infiltration of macrophages, epithelioid macrophages, lymphocytes, plasma cells and multinucleated giant cells in the periphery. These areas were surrounded by significant proliferation of fibrous connective tissue that was also infiltrated by macrophages, lymphocytes and plasma cells. The epidermis showed moderate diffuse acanthosis and moderate diffuse parakeratotic hyperkeratosis. In the lymph nodes, within the nodal parenchyma, there were multifocal-to-coalescing areas with marked caseous necrosis, mineralization and significant inflammatory infiltration of macrophages, epithelioid macrophages, lymphocytes, plasma cells and multinucleated giant cells. No acid-fast resistant bacilli or fungal structures were observed in the Ziehl–Neelsen or Grocott methenamine silver histochemical analyses, respectively.

## 3. Discussion

The diagnosis of *Pelodera* spp. dermatitis in a buffalo from Marajó Island, state of Pará, was confirmed by histopathological lesions of the skin associated with the presence of intralesional rhabditiform larvae, morphologically consistent with *Pelodera* spp. There are still no reports of granulomatous dermatitis caused by *Pelodera* spp. in buffalo; our hypothesis is that some risk factors may be contributing to the occurrence of this disease in buffalo on Marajó Island.

According to Saari and Nikander [7], *Pelodera* spp. dermatitis in dogs is often associated with direct contact with contaminated wet straw bedding or hay, an environment similar to that which is found at the site where the disease occurred on Marajó Island. These conditions allow *Pelodera* spp., a free-living saprophytic nematode, to complete its life cycle. The contact between this agent and the buffalo probably occurred when the animal was seeking to ease the heat during the hottest times of the year in these humid areas, which are rich in decomposing materials. According to Damasceno et al. [2], unlike in cattle, thermoregulation in buffalo is less efficient and the animals require these environments, which favor greater exposure to the parasite, for heat exchange.

Chronicity of the disease in the animal was determined by the severe clinical signs and the clinical history provided by the caretaker, who mentioned that the animal had been showing growths on the skin for two years, which were also observed on clinical examination at the time of the technical visit. Despite a lesion on the hoof and affected bones and tendons, as seen on clinical examination and confirmed by necropsy, the animal did not claudicate, unlike buffalo with similar bone lesions and like in cases of iron deficiency, in which there is fracture of the phalanges, formation of osteophytes, and bone fragility [4,20], and in buffalo with degenerative joint disease (DJD) [21].

The macroscopic skin lesions characterized by nodules of varying sizes in the buffalo of the present study differed from those observed in other animals and in humans [7,8,9,10,11,12,13,14,15,16,17,18,19], which are characterized by erythema, alopecia, flaking, scabs and occasionally pustules. In this buffalo, the nodules probably originated from the pustules formed by the activity of the larvae on the skin. It is of note that the presence of lesions in the nostrils and in vaginal mucosa does not rule out that these sites were also an entry for the parasite.

Encapsulation of the granulomas by fibrous connective tissue probably occurred to isolate the caseous content present inside this structure, as occurs in cases of tuberculosis [22] and other granulomatous skin diseases. However, it is possible that the infectious agents were carried by larvae through the skin, like in other species of animals [12]. The ultrasound findings were consistent with the nature of the content of the granulomas [23].

The lesions on the macerated bones, such as the osteophytic proliferation with a rough and irregular appearance, can be explained by the stimuli caused by the inflammatory process in the periosteum. This bone change was possibly accelerated, not only by the pressure of the granulomas on the surface of the bone, but also by phosphorus deficiency, since the animals did not receive mineral supplementation and it is a common deficiency on Marajó Island [4,20]. The concavities/depressions seen on the diaphysis may have been caused by the pressure exerted by the granulomas on the bone surfaces throughout the years.

Histopathology showed a reduced number of hair follicles, larvae on the hair follicles, an acanthotic epidermis with parakeratotic hyperkeratosis and an inflammatory infiltrate composed of macrophages, epithelioid macrophages, lymphocytes, plasma cells and multinucleated giant cells, similar to the histopathological findings reported by other authors in different species of animals with *Pelodera* dermatitis [7,8,10,11,13,14,15,16,17].

Because of the predominance of granulomas found during the necropsy, it was necessary to make the differential diagnosis between this disease and, above all, tuberculosis, which also progresses with the formation of granulomas; however, this disease was ruled out after Ziehl–Neelsen staining of skin and lymph node samples. Other findings that contributed to the exclusion of this disease were the absence of lesions in the internal organs and lymph nodes and the absence of studies describing cutaneous tuberculosis in buffalo.

A differential diagnosis was also carried out with multicentric lymphoma in buffaloes, which progresses with an increase in volume of the skin and subcutaneous tissue, which, at necropsy, are characterized by whitish nodules with varying sizes and a soft consistency. A histological examination revealed the presence of tumor masses, with proliferation of round, discreetly pleomorphic cells with little cytoplasm, large nuclei and a rounded ovoid basophilic shape, sometimes with loose chromatin and prominent nucleoli, compatible with lymphocytes [24]. This is unlike parasitic granulomatous dermatitis caused by *Pelodera* spp., which macroscopically is characterized by caseous, coalescent and yellowish cutaneous granulomas surrounded by fibrous connective tissue and also by the presence of the parasite inside these granulomas observed via skin histopathology.

The exclusion of data referring to buffaloes treated by veterinary colleagues with clinical signs similar to the disease in the same region as our study was due to the absence of complementary tests that would confirm the presence of *Pelodera* spp. in these animals; however, they are epidemiological data of interest for future investigations.

The control of this disease on Marajó Island, State of Pará, will probably be difficult due to the extensive breeding system used in the majority of the region’s buffalo breeding farms, and will be even more difficult due to the climatic/environmental characteristics of the island, namely, the periods of intense rainfall and prolonged drought, and the buffalo’s physiological particularities, such as less efficient body temperature regulation, which causes these animals to gather near lakes during periods of drought to regulate their body temperature. Further studies are recommended to assess the incidence of this parasite in buffalo on Marajó Island, State of Pará, as well as the coexistence of infectious agents that worsen this parasitosis in buffalo.

## 4. Conclusions

This is the first report of parasitic granulomatous dermatitis in buffalo caused by *Pelodera* spp. in Brazil, in the Amazon region. The severity of the lesions was due to the fact that the animal had the disease for a long period, and the histopathological findings effectively confirmed the diagnosis.

## Figures and Tables

**Figure 1 animals-14-01328-f001:**
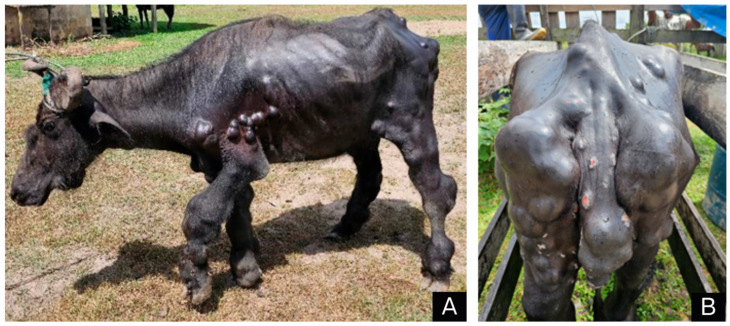
Parasitic granulomatous dermatitis caused by *Pelodera* spp. in buffalo; (**A**) multifocal to coalescing, well-delimited and sometimes ulcerated nodular skin lesions of varying size were distributed across several regions of the body; (**B**) same aspect of the lesions with an emphasis on the posterior region.

**Figure 2 animals-14-01328-f002:**
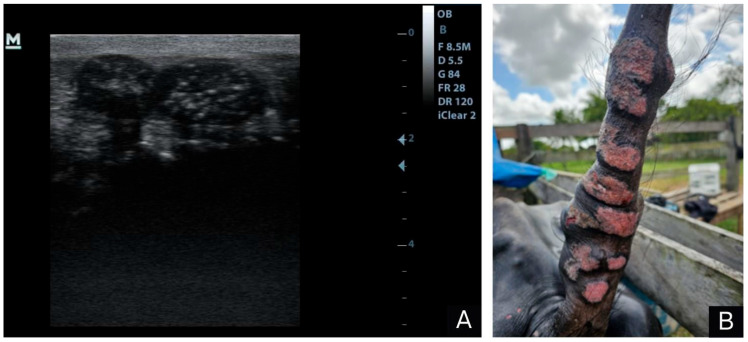
Parasitic granulomatous dermatitis caused by *Pelodera* spp. in buffalo; (**A**) ultrasound image of skin nodules characterized by ovoid structures located in the subcutaneous tissue, with a multifocal and sometimes coalescent distribution delimited by a capsule. Sometimes, these nodules were thicker and sometimes they were delimited by a thin hyperechoic line, with heterogeneous contents in echogenic areas and hypoechogenic liquid content tending to anechoic in shapeless hyperechoic areas. (**B**) Ulcerated, well-delimited, multifocal to coalescing skin lesions with irregular borders and of varying sizes distributed along the ventral region of the tail.

**Figure 3 animals-14-01328-f003:**
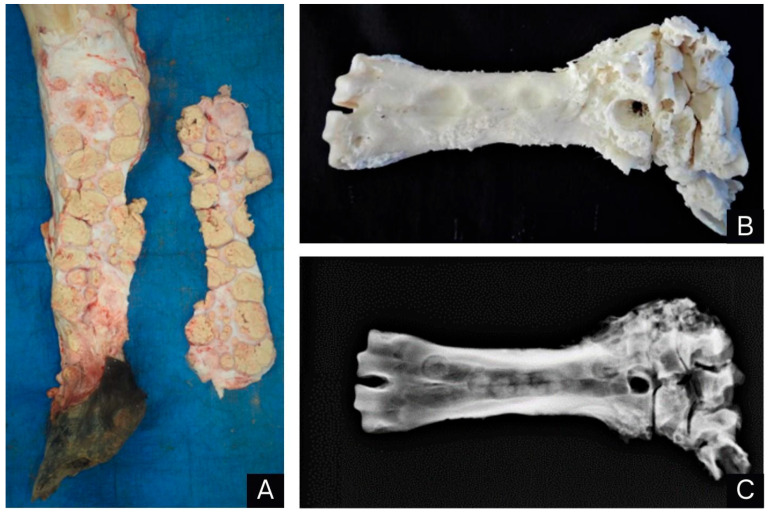
Parasitic granulomatous dermatitis caused by *Pelodera* spp. in buffalo: (**A**) cut surface of the multifocal to coalescing, yellowish, caseous skin granulomas, surrounded by fibrous connective tissue. (**B**) Osteophytic proliferation of rough-to-irregular appearance on the periosteal surface of the metacarpal and on the carpal bones after maceration. Concavities where the granulomas exerted more pressure are seen on the caudomedial side of the diaphysis. (**C**) X-ray image showing mild spiculate periosteal reactions, mainly on the proximal third of the metacarpal and on the bones of the adjacent carpal. Medullary canal of the metacarpal with a heterogeneous appearance and radiopaque circumscribed areas. A large area of osteolysis is seen on the proximal third of the metacarpal. Reduced intracarpal and carpometacarpal joint space.

**Figure 4 animals-14-01328-f004:**
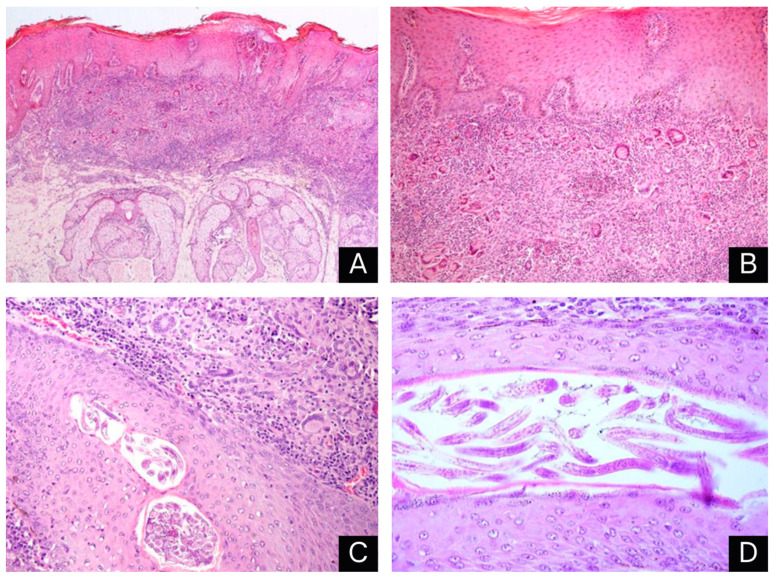
Parasitic granulomatous dermatitis caused by *Pelodera* spp. in buffalo: (**A**) Inflammatory infiltrate in the superficial dermis, absence of hair follicles and evident acanthosis and epidermal hyperkeratosis. HE, 40×. (**B**) Inflammatory infiltrate rich in multinucleated giant cells in the dermis. HE, 100×. (**C**) Granulomatous inflammation around a hair follicle full of parasitic structures. HE, 200×. (**D**) Several sections of rhabditiform parasites inside the hair follicle. HE, 400×.

## Data Availability

Data are contained within the article.

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
