# Peer review of "Parasitic Granulomatous Dermatitis Caused by Pelodera spp. in Buffalo on Marajó Island, Pará"

_animals, 2024, doi:10.3390/ani14091328_

Round 1
Reviewer 1 Report
Comments and Suggestions for Authors
Abstract
In the first sentence I suggest separating the sentences by putting first: “This is the first report of parasitic granulomatous dermatitis caused by Pelodera spp. in a buffalo” and subsequently address the characteristics of the affected animal and its location: The affected animal was a “seven-year-old female buffalo of the Murrah breed and belonging to a property located in the Marajó Island, in the State of Pará.”
2. Case report (Suggestions)
First paragraph (lines 62 to 67): There is only one case in how many buffaloes are in the herd? Is there any reproductive and productive history?
Second paragraph (lines 73 and 74): the ultrasound description of the lesions is very simple, Furthermore, granulomatous lesions usually present a hyperechoic and heterogeneous appearance well delimited but without large fibrous capsule, and may or may not present areas of acoustic shadow due to mineral deposition or anaechoic areas (caseous necrosis) when they are pyogranulomatous lesions and not just granulomatous lesions. There is no ultrasound image with a detailed description in the body of the report. Despite being a non-specific diagnostic technique, the correct description of the lesions and the exposure of at least one image of one of the lesions would enrich the case report.
P.S.: during de hystophatology description the authors afirm that are “Amidst this infiltrate there were areas of multifocal necrosis with significant deposition of cell debris”(lines 109 and 110), its possible that this caracterizes a pyogranuloma?
Figure 2, c (Suggestion)
In the description revove the word “OSSO 1” of the foto.
3. Discussion
Question 1 (lines 142 to 144): Do the authors have any hypothesis that could justify the occurrence of the disease only in the animal in question since other animals from the same herd must have had access to the same humid place infested by Pelodera spp.?
Suggestion 1 (lines 164 and 165): Add a reference to an ultrasound description of the granuloma after the statement.
Suggestion 2 (lines 166 and 170): It would be interesting to discuss these statements with authors who found similar findings even in different diseases or who address aspects of bone metabolism and possible chronic lesions in these organs.
Suggestion 3 (lines 176 and 181): The statements in this paragraph are correct, however the authors did not mention other authors who support their statements.
Question 2 (lines 182 to 190): If this disease has the potential to be so important for Buffalo breeding on Marajó Island, why was only one animal diagnosed and not a group of animals?
Suggestion 4 (lines 182 and 190): This paragraph seems like a conclusion and I was unable to identify a discussion with other works.
4. Conclusion
Question 2 (lines 192 to 195): Ultrasound could be used as a tool for differential diagnosis between parasitic granulomatous dermatitis in buffalo caused by Pelodera spp. and chronic tuberculosis because in chronic tuberculosis there are granulomatous lesions in abdominal and/or thoracic organs?
Reviewer 2 Report
Comments and Suggestions for Authors
Dear authors,
I would like to congrats you for this very nice piece of paper. I liked very well the work.
I have no major comments.
All the best

Author Response
Thank you very much for your considerations.Reviewer 3 Report
Comments and Suggestions for Authors
The topic is very relevant, the authors making the first report of parasitic granulomatous dermatitis caused by Pelodera spp. in a female buffalo in Brasil
The methodology is modern and complex, using paraclinical tools, especially histopathology and special bacterial stains.
The results have revealed dermic inflammatory infiltrate consisting of macrophages, epithelioid macrophages, lymphocytes, plasma cells, and multinucleated giant cells, multifocal-to-coalescent. Numerous cross and longitudinal sections of small rhabditoid nematodes characterized by a thin cuticle, platymyarian musculature, intestinal tract, rhabditiform esophagus, and lateral alae, morphologically compatible with Pelodera spp. were identified.
The conclusions are consistent with the evidence and arguments presented.
I suggest some corrections
1. References could be improved. In Introduction you may refer also to
Acatrinei, D.; Miron, L.; Solcan, G.; 2006, Pelodera strongyloides dermatitis in dog - case report. Revista Română de Medicină Veterinară, 2006, 16, 1, pp 95-100
2. Lines 179-181 authors mention. Other findings that contributed to the exclusion of this disease were the absence of lesions in the internal organs and lymph nodes and the absence of studies describing tuberculosis in buffalo. I don t agree
See Radostits, O. M.; Blood, D. C.; Gay, J., Hinchcliff, K.W., 2007, Veterinary Medicine A Textbook of the Diseases of Cattle, Sheep, Pigs, Goats and Horses. 10th Edition, W.B. Saunders Company Ltd., London, chap 19, pag 1007 and you ll find out that buffaloes are even reservoir for tuberculosis in Australia and South Africa
Reviewer 4 Report
Comments and Suggestions for Authors
This report describes the first case of Pelodera dermatitis in a buffalo in a particular region of Brazil. The case presented is very severe. The authors extensively describe the clinical signs, gross lesions, radiographic findings and histopathological findings. The report is very well written and thorough. The images are of high quality and the text and legend accurately describe the photos and photomicrographs. The gross and histopathological descriptions are very well written and the microscopic description of the parasite is very thorough.
Minor findings:
In line 45, the authors include hot-iron marking in a list of diseases that affect buffalo in this regions. Is this considered a disease?
Line 181: The authors rule out tuberculosis based on several criteria, the last of which is the absence of studies describing tuberculosis in buffalo. I am not sure this is a reason to rule out tuberculosis. The lack of published information may not be due to the fact that it does not occur.
